# Using a Deep Learning Model to Address Interobserver Variability in the Evaluation of Ulcerative Colitis (UC) Severity

**DOI:** 10.3390/jpm13111584

**Published:** 2023-11-08

**Authors:** Jeong-Heon Kim, A Reum Choe, Yehyun Park, Eun-Mi Song, Ju-Ran Byun, Min-Sun Cho, Youngeun Yoo, Rena Lee, Jin-Sung Kim, So-Hyun Ahn, Sung-Ae Jung

**Affiliations:** 1Department of Medicine, Yonsei University College of Medicine, Seoul 03722, Republic of Korea; air2kim431@yuhs.ac (J.-H.K.);; 2Medical Physics and Biomedical Engineering Lab (MPBEL), Yonsei University College of Medicine, Seoul 03722, Republic of Korea; 3Department of Radiation Oncology, Yonsei Cancer Center, Heavy Ion Therapy Research Institute, Yonsei University College of Medicine, Seoul 03722, Republic of Korea; 4Department of Internal Medicine, Ewha Womans University College of Medicine, Seoul 03760, Republic of Korea; archoi20@ewha.ac.kr (A.R.C.); splendidyh1029@ewha.ac.kr (Y.P.);; 5Department of Pathology, Ewha Womans University College of Medicine, Seoul 03760, Republic of Koreayeyoo@ewha.ac.kr (Y.Y.); 6Department of Bioengineering, Ewha Womans University College of Medicine, Seoul 03760, Republic of Korea; 7Ewha Medical Research Institute, Ewha Womans University College of Medicine, Seoul 03760, Republic of Korea

**Keywords:** endoscopy, ulcerative colitis, deep learning, interobserver variation, severity

## Abstract

The use of endoscopic images for the accurate assessment of ulcerative colitis (UC) severity is crucial to determining appropriate treatment. However, experts may interpret these images differently, leading to inconsistent diagnoses. This study aims to address the issue by introducing a standardization method based on deep learning. We collected 254 rectal endoscopic images from 115 patients with UC, and five experts in endoscopic image interpretation assigned classification labels based on the Ulcerative Colitis Endoscopic Index of Severity (UCEIS) scoring system. Interobserver variance analysis of the five experts yielded an intraclass correlation coefficient of 0.8431 for UCEIS scores and a kappa coefficient of 0.4916 when the UCEIS scores were transformed into UC severity measures. To establish a consensus, we created a model that considered only the images and labels on which more than half of the experts agreed. This consensus model achieved an accuracy of 0.94 when tested with 50 images. Compared with models trained from individual expert labels, the consensus model demonstrated the most reliable prediction results.

## 1. Introduction

Ulcerative colitis (UC) is an idiopathic, chronic inflammatory disease of the colon mucosa, usually beginning in the rectum and extending proximally through all or part of the colon [1]. With alternating periods of exacerbation and remission, the clinical course is unpredictable. Diagnosis of UC is based on clinical symptoms and confirmed by objective findings from endoscopic and histological examinations. Endoscopic evidence of UC includes persistent colonic inflammation, with confirmatory biopsy specimens indicating chronic colitis [2]. The endoscopy plays an important role in managing patients with UC, allowing us to visualize and assess disease severity. Consequently, the objective assessment provided by endoscopy is important to the optimal management of patients with UC.

The most widely used scoring systems for assessing endoscopic disease activity in UC are the Mayo endoscopic subscore and the Ulcerative Colitis Endoscopic Index of Severity (UCEIS) [3]. The UCEIS, which was developed using a linear mixed regression model, assesses the extent of endoscopic severity based on three variables: vascular pattern (normal, 1; patchy obliteration, 2; or obliterated, 3), bleeding (none, 1; mucosal, 2; luminal mild, 3; or luminal moderate or severe, 4) and erosions and ulcers (none, 1; erosions, 2; superficial ulcer, 3; or deep ulcer, 4) [4]. It is challenging to correctly grade colonoscopies using the UCEIS, with even experienced and sufficiently trained experts showing interobserver variability. As a result, researchers have been working to develop a deep learning system for the consistent and objective analysis of UC endoscopic images based on artificial intelligence (AI) [5,6,7,8,9,10].

Our study used AI for the endoscopic evaluation and diagnosis of patients with UC. AI was first used in 2003 to assess endoscopic severity in UC patients. Sasaki et al. defined the Matts score for grading endoscopic severity using pictorial parameters of mucosal redness from 133 digital colonoscopy fixed images of 55 patients with UC. The degree of mucosal redness was measured as a hemoglobin index through a Bayesian-driven computer-aided detection algorithm. This algorithm could differentiate the Matts grades based on the kurtosis of hemoglobin index with high sensitivity and specificity [5]. More recently, Ozawa et al. attempted to detect mucosal remission or activity in UC patients using AI and a computer-aided detection system based on convolutional neural networks (CNN) and trained on large datasets of endoscopic still images. The system showed a high level of performance, with areas under the receiver operating characteristic curve of 0.86 and 0.98 to identify Mayo 0 and 0–1, respectively [6].

In addition, many other authors have conducted UC-related research using AI. Sutton et al. utilized AI to differentiate UC from other intestinal diseases and to assess the severity of UC endoscopic ulcers, achieving an accuracy of 87.50% and an area under the curve of 0.90 with 851 images from UC patients [7]. Takenaka et al. created a deep neural network system that analyzed endoscopic images from 2012 UC patients (totaling 40,758 images) and 6885 biopsy outcomes, achieving 90.1% accuracy in identifying remission in endoscopy [8]. Yao et al. trialed a fully automated video system for analyzing and grading endoscopic disease in UC. The system, working with videos of the clinical trial set comprising 51 high resolution images and 264 tests, correctly differentiated between remission and active disease in 83.7% of cases [9]. Gottlieb et al. verified a deep learning algorithm’s capability to predict levels of UC severity from full-length endoscopy videos from 249 patients, displaying an area under the curve ranging from 0.787 to 0.901 for the endoscopic Mayo Score and 0.855 for the UCEIS [10]. Finally, Bossuyt et al. developed an operator-independent computer-based tool to determine UC activity based on endoscopic images from 29 consecutive UC patients and 6 healthy controls. This tool’s readings correlated significantly with the Robarts histological index, Mayo Endoscopic Score, and UCEIS. These studies collectively indicate the growing potential of AI in improving the diagnosis, assessment, and management of UC [11].

Therefore, AI implementation in UC is promising for improving the assessment of disease activity and reducing interobserver variability in grading such activity. In most studies, the primary focus has been on the binary classification of Ulcerative Colitis (UC) states, differentiating between the inactive and active phases of the disease.

In this study, we developed a model that predicts three stages of UC severity in the diagnosis of endoscopic images from patients with UC. Furthermore, to enhance the objectivity and precision of UC diagnosis, we constructed a robust deep learning model that effectively reduces discrepancies between different expert evaluations.

## 2. Materials and Methods

In this study, we aimed to analyze statistically experts’ scoring differences for patients with UC and assess quantitatively the impact of this interobserver variability on diagnostic outcomes [12,13,14]. To achieve this, we trained a deep learning network called consensus data using only images for which expert scoring was consistent. We compared the performance of the deep learning model, which was trained using each expert’s scoring, with the performance of different models trained to diagnose test images. Figure 1 shows the flowchart of this study.

### 2.1. Patients and Images

A total of 254 rectal endoscopic images obtained from 115 patients with ulcerative colitis who underwent endoscopy at Ewha Womans University Seoul Hospital between 10 June 2019, and 29 February 2021, were targeted (Table 1). Patients with Crohn’s disease, resection before the colonoscopy date, or other bowel resection were excluded from the study. The study protocol was approved by the Ethics Committee of Ewha Womans University Seoul Hospital (IRB no. 2023-03-028).

The severity of ulcerative colitis was classified into three categories: remission/mild, moderate, and severe. These labels were used as input data for the deep learning network models. Figure 2 shows the collected endoscopic images and labels in this study.

The endoscopic images were captured using a CV-290 (Olympus, Tokyo, Japan). They are RGB images with resolutions of 543 × 475 and 1242 × 1079, and an 8-bit color depth.

### 2.2. Scoring System

We introduced the consensus approach in this study. Under this approach, five experts independently assign scores, but the final score is only adopted if at least three experts assign the same score, termed “consensus data”. This method minimizes bias in evaluations from individual experts’ judgments, contributing to increasing reliability.

### 2.3. Deep Learning Network

For image classification, we selected 13 deep learning network models based on CNN (Figure 3), which are known for their exceptional performance in this field: DenseNet-121, MobileNetV2, DenseNet201, InceptionV3, EfficientNetB0, EfficientNetB7, MobileNetV3Large, ResNet152V2, ResNet50, ResNet50V2, VGG19, VGG16, and Xception.

These models were chosen based on their exceptional performance in this field and their extensive application in research. Despite their diverse architectures, all models utilize convolutional and pooling layers for feature extraction and dimensionality reduction, which makes them highly efficient for high-level image classification tasks, aligning perfectly with the requirements of our study.

All of these models are TensorFlow implementations initialized using weights from the ImageNet dataset [15]. This method, known as transfer learning, is common practice in many fields of medical imaging and has proven to be exceptionally successful [16]. The principal advantage of this approach is that it uses the pre-learned weights from the lower layers of these models. These lower layers often detect more generalized features, such as edges or textures, which are universal to many image classification problems. Our study employed end-to-end training, achieving satisfying results. However, for cases where this approach is not as effective, ‘fine-tuning’—freezing the lower layers—could be a viable alternative.

Our initial dataset consisted of endoscopic images split into training and test sets at an approximate ratio of 8:2 (Table 2). To further enhance and generalize our deep learning model, we supplemented our dataset with images from the HyperKvasir open dataset, obtained from Bærum Hospital between 2008 and 2016, which boasts a collection of 110,079 images. From this, 10,662 images were labeled across 23 classes of finding [17], with a significant portion related to pathological findings like Barrett’s esophagus, esophagitis, polyps, ulcerative colitis, and hemorrhoids. From the 851 UC images, we carefully selected those with quality endoscopic features. Five experts then assessed these images for severity, resulting in a consensus on 267 images. These were combined with our initial 254 images, culminating in a total dataset of 521 images for deep learning training.

The model training was conducted for 30 epochs, with a batch size of 30. The categorical cross-entropy loss function, commonly used for multiclass classification, was selected, and the Adam optimization algorithm was used. The learning rate was set to 1 × 10^−4^, and all images were downscaled to a size of 543 × 475.

Accuracy, recall, precision, and F1 score were utilized to assess the performance of our deep learning model comprehensively. These metrics are used to evaluate the model’s classification performance quantitatively, also reflecting the approach’s sensitivity and the harmonic mean of the metrics.

### 2.4. Data Preprocessing

Endoscopic images inherently possess characteristics such as reflections caused by the light source and dark regions where the light does not reach. These phenomena exist as artifacts in deep learning training, and it is crucial to control them to ensure the model’s accuracy and reliability [2,3]. In order to eliminate such artifacts, we attempted to remove the areas corresponding to light reflection in the RGB channels. However, we faced challenges in accurately isolating only the areas of light reflection, as parts of ulcer or erosion regions were also eliminated alongside the reflective regions in the RGB channels. This complexity posed a difficulty in precisely detecting the areas of light reflection (Figure 4).

In the data preprocessing stage, the color space of the images was converted from RGB to HSV to eliminate reflections and dark areas (Figure 5) [18,19]. The ranges to detect reflections and dark areas were (0, 360) for H, (90, 255) for S, and (65, 236) for V.

With this method, the identified regions were converted into binary mask images, which were then multiplied with the original RGB endoscopic images. Subsequently, the empty spaces were filled using an inpainting technique [20].

To enhance the generalization performance of the model through data augmentation, the following techniques were applied: rotation range (360 degrees), zoom range (15%), width shift range (20%), height shift range (20%), shear range (15%), horizontal flipping, and filling mode (“reflect”).

### 2.5. Interobserver Variation

We employed the intraclass correlation coefficient (ICC) to assess the agreement among UCEIS scores (ranging from 0 to 8) assigned by the experts (Table 3). The ICC is a statistical methodology that measures the level of agreement among observations by calculating the ratio of between-observer variance to the total variance among observations. Through this approach, we evaluated the consistency of scores assigned by multiple experts to the same images.

We used Fleiss’ kappa index to evaluate the agreement in severity classification based on the UCEIS assigned by the experts (Table 4). In this context, the labels were classified into four categories: remission, mild, moderate, or severe. Fleiss’ kappa is a suitable statistical method for measuring agreement among evaluators assessing categorical data.

## 3. Results

### 3.1. UCEIS Score Estimation

To measure the interobserver variance, we closely examined the UCEIS scores and severities provided by each expert. The UCEIS scores and severities evaluated by five experts and the consensus data are shown in Figure 6a,b. When we examined the UCEIS scores given by each expert, we observed a maximum difference of 63 images. Of the total 254 images, we constructed a consensus data set using 220 images that were scored identically by more than half of the experts. Similarly, when we considered the severity of the condition, we found a maximum difference of 79 images. In the consensus data criteria, out of 254 images, all the images met the consensus conditions.

### 3.2. Statistical Analysis of Interobserver Variance

The interobserver variance among five experts was assessed based on UCEIS scores. The ICC, a metric of interobserver consistency, was 0.8431, indicating good agreement among the observers’ assessments [21].

Subsequently, UCEIS scores were transformed into measures of UC severity, and interobserver variance was recalculated among the same set of experts. In this context, the calculated kappa coefficient was 0.4916. While this value does not imply perfect agreement among the observers, it does denote moderate agreement, demonstrating a certain degree of uniformity in the classification of UC severity [21].

### 3.3. Outcome of the Deep Learning Network Model

We evaluated the performance of 13 deep learning network models of consensus data, which is crucial for ensuring reliability in endoscopic image classification. Table 5 presents the results, including accuracy, F1 score, recall, and precision. The models are listed in descending order based on accuracy. EfficientNetB0 exhibited the highest overall performance. With an impressive accuracy of 79.20%, coupled with a balanced F1 score of 81.25%, this model demonstrated robust capabilities in the classification tasks. The confusion matrix for the EfficientNetB0 model shows that it correctly identified remission and mild 39 times out of 51, and moderate 39 times out of 41 (Figure 7). For severe cases, the model correctly identified 9 out of 12. The individual deep learning results for each of the five experts are presented in Appendix A.

### 3.4. Differences in Accuracy for Each Model

We developed six deep learning network models, trained with labels from five experts as well as consensus data, without using the HyperKvasir dataset. Table 6 shows the number of pass/fail images when labels were predicted for 50 test images. Expert A and the consensus networks predicted the most accurately.

## 4. Discussion

This study aimed to measure and quantify the differences in interpretation among experts analyzing endoscopic images of patients with UC. A deep learning model using images and labels that were agreed upon by more than half of the participating experts was developed. The proposed consensus approach can effectively reduce variations in expert opinions while preserving the diagnostic patterns specific to each institution.

Interobserver variance in both UCEIS scores and UC severity offers important insights into the complexities of medical diagnosis and evaluation. While the ICC for UCEIS scores stands at 0.8431, indicating a good level of agreement among experts, the kappa coefficient for UC severity shows a value of 0.4916, signifying only a moderate consensus. It’s important to recognize that clinical decisions, especially therapeutic choices, are primarily grounded on the assessment of UC severity rather than just UCEIS scores. This underscores the profound significance of the observed interobserver variation and points to the existence of such variances in real-world clinical settings. The difference in these evaluations emphasizes the need for a deep learning approach to assessing severity, ensuring consistency in patient care and treatment decisions.

The implementation of our deep learning model offers a transformative approach to diagnostic procedures in gastroenterology, particularly for those related to UC. By providing a standardized diagnostic guide, it serves as an invaluable asset not only for fellows or beginners entering the clinical settings but also for seasoned practitioners handling many cases on a day-to-day basis. This standardized approach is especially significant considering the inherent subjectivity and variability in interpreting endoscopic images. As individual reading tendencies and biases might evolve with experience, the model acts as a consistent anchor, mitigating the risk of divergent interpretations. Furthermore, the model’s adaptability ensures it remains relevant and updated, reflecting advances in understanding and shifts in diagnostic criteria.

The comparison with previous studies on the evaluation of UC severity using deep learning is presented in Table 7. One of the notable features of our study is the advanced technology for handling artifacts. Using this method, we effectively eliminated light reflections. Further adjustments to the HSV range made it possible to fully detect even the dark areas that aren’t directly illuminated. However, when we tried applying inpainting techniques afterward, we experienced a significant loss in crucial endoscopic data information. As a result, we decided to optimize the HSV range to appropriately capture the dark areas without compromising on data quality. This thoughtful calibration and approach substantially boosted the performance of our deep learning model. Moreover, the number of images used in our study might be relatively small when compared with other deep learning studies involving endoscopic images; however, while the volume of data is important in deep learning, we recognized that the use of high-quality data is even more critical [22,23,24]. Therefore, we selectively utilized images of the rectum from patients. This is because UC symptoms first appear in the rectum, manifesting most prominently there before gradually spreading to other areas.

Additionally, in the labeling process, we used only data that met certain criteria to minimize interobserver variance. This approach ensured that we maintained a high standard of data quality throughout the study. When labels were consistent across experts, it indicated that the chosen images were both representative and clear, eliminating potential ambiguities in interpretation. Still, in situations where data was scarce, we combined our data with the HyperKvasir dataset for deep learning models. Incorporating external datasets is a common practice to enhance the robustness of models, especially when native datasets might not provide sufficient variability. From the HyperKvasir dataset, we selectively curated images that were suitable for endoscopic judgment and of good quality to form a consensus data set. This curation process was rigorous, ensuring that only the most pertinent and clear images were added to our data pool. Owing to our meticulous data management, combined with a comprehensive selection process, we achieved excellent accuracy even with a relatively small dataset [25,26].

Nevertheless, we acknowledge the lack of data for severe cases in our study. This limited number of severe images affected the overall performance of our model. This discrepancy arose primarily because there were significantly fewer images of severe cases compared to those of remission/mild or moderate severity. Such imbalances in the dataset can introduce biases into deep learning models, affecting their generalizability in diverse clinical settings. Understanding this limitation, we emphasize the urgent need to gather a more extensive collection of image data, particularly those depicting severe conditions, to strengthen the predictive capabilities of our model. To address this gap, we are laying the groundwork for a multicenter study in the imminent future. Collaborating with various centers will not only grant access to a larger dataset but also ensure its diversity, covering a broader range of clinical scenarios. Through these efforts, we aim to fine-tune our model, striving to enhance its reliability across the spectrum of UC severity.

The decision to combine the remission and mild categories in the dataset used to train the deep learning model is grounded in clinical rationale and informed by treatment objectives. From a treatment perspective, patients in remission or with mild symptoms are often not the primary targets for aggressive intervention. The combination of these categories into a single group reflects a clinically meaningful distinction in the condition’s management. This approach ensures that the model’s predictions align more closely with the clinical considerations that guide treatment decisions, potentially improving the model’s utility in the real-world clinical setting.

We also collected pathologic readings of endoscopic images for our study and sought the input of two pathologists to ensure reliability. In cases where their opinions diverged, the pathologists engaged in discussions to reach a consensus on the pathological interpretation [27,28]. Figure 8 illustrates the distribution of severity based on both pathology and clinical findings. The differences between the pathological and clinical findings are thought to arise because the pathological findings are evaluated only in the biopsy tissue, while the clinical findings are evaluated in the entire endoscopic image. In clinical practice, biopsy results are important, but decisions are also influenced by the size and appearance of lesions visible on endoscopic images [27]. As a result, we chose not to utilize the pathology findings for data labeling in this study, focusing instead on other significant factors.

**Table 7 jpm-13-01584-t007:** Comparison of previous studies and our study.

Study (Year)	Data Set	Outcome	UC Severity Estimation
Ozawa et al. (2019) [6]	26,304 images/444 patients	MES (Mayo 0, Mayo 1, and Mayo 2–3)	Accuracy: 0.704
Stidham et al. (2019) [29]	14,862 images/2778 patiens	MES	Kappa: 0.840 Accuracy: 0.778
Maeda et al. (2019) [30]	12,900 images/87 patients	Histologic inflammation estimation (active vs. healing) MES 0 vs. MES 1	Accuracy: 0.910 Sensitivity: 0.650 Specificity: 0.980
Bhambhvani et al. (2020) [31]	90% of 777 images/777 patients	MES estimation (Mayo 1, Mayo 2, and Mayo 3)	Accuracy: 0.772 Sensitivity: 0.724 Specificity: 0.857
Gottlieb et al. (2021) [10]	80% of 795 videos/249 patients	MES UCEIS	QWK: 0.844 Accuracy: 0.702 Sensitivity: 0.716 Specificity: 0.901
Yao et al. (2021) [9]	16,000 images/3000 patiens	MES	QWK (A): 0.840 Accuracy (A): 0.780 QWK (B): 0.590 F1 (B): 0.571
Schwab et al. (2022) [32]	80% 1881 videos/726 patiets	MES	QWK: 0.680 (video level) QWK: 0.660 (frame level)
Luo et al. (2022) [33]	(A): 80% of 9928 images, (B): 80% of 4378 images /1317 patients	MES	Accuracy (A): 0.906 F1 (A): 0.868 Accuracy (B): 0.916 F1 (B): 0.858
Polat et al. (2023) [34]	9590 images/462 patients	MES	QWK: 0.854 F1: 0.697 Accuracy: 0.772 Sensitivity: 0.693 Specificity: 0.911
Kim et al. (Our study)	(A): 254 images/115 patients (B): 262 images	UCEIS	Accuracy: 0.792 F1: 0.760

MES: Mayo endoscopic sub-score.

## 5. Conclusions

In this study, we developed a consensus model for reliably interpreting endoscopic images. To do so, we gathered label data created from the collective opinions of multiple experts and then evaluated the accuracy of the model. The results of our study are significant as they propose a method to reduce differences and variations that individual experts may introduce. By adopting a consensus approach, we can improve the consistency and reliability of interpreting endoscopic images.

## Figures and Tables

**Figure 1 jpm-13-01584-f001:**
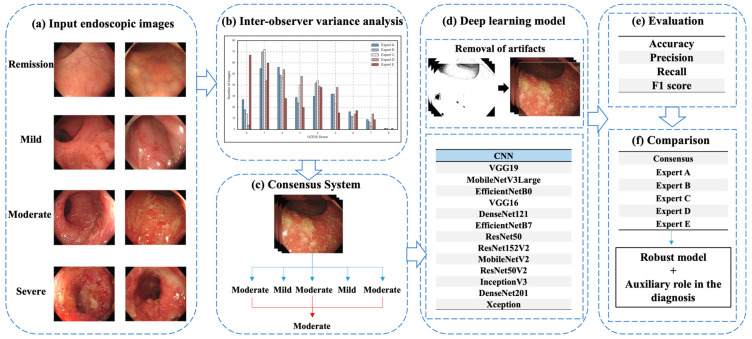
Study flowchart.

**Figure 2 jpm-13-01584-f002:**
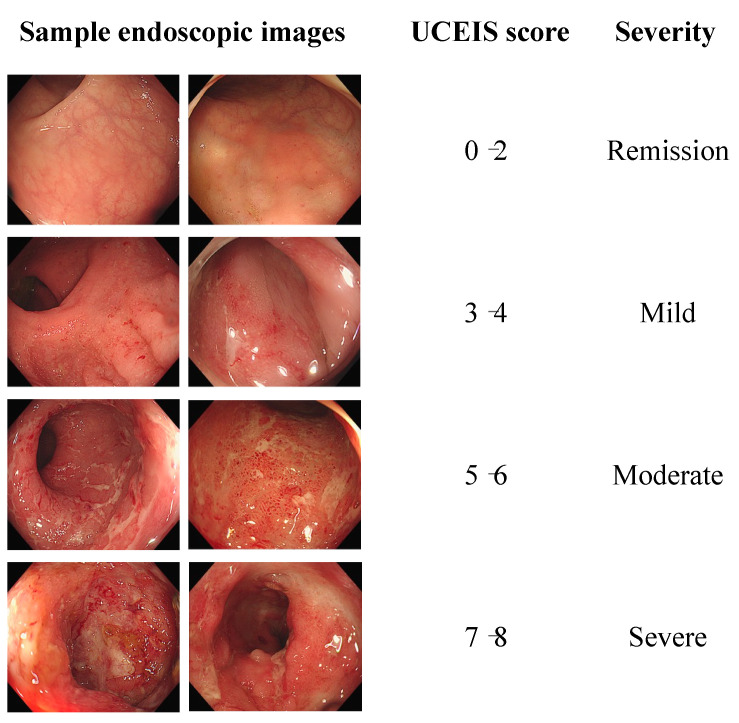
Endoscopic images and ulcerative colitis labels determined using the Ulcerative Colitis Endoscopic Index of Severity (UCEIS).

**Figure 3 jpm-13-01584-f003:**
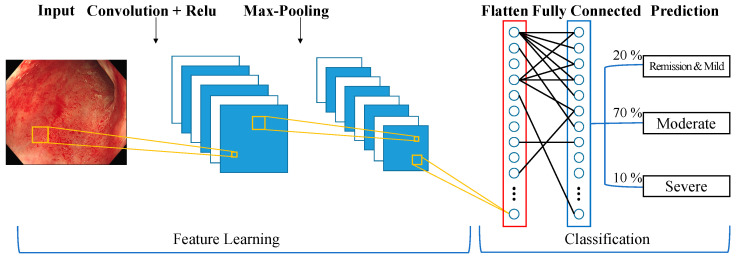
Convolutional neural network architecture.

**Figure 4 jpm-13-01584-f004:**
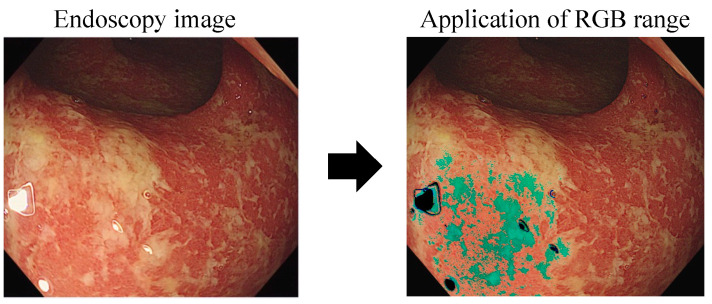
S Specifying the range for RGB channels to eliminate light reflection: Red channel (0, 210), Green channel (0, 210), Blue channel (0, 210), White: light reflection, Black: area corresponding to light reflection, Green: areas representing ulcer or erosion.

**Figure 5 jpm-13-01584-f005:**
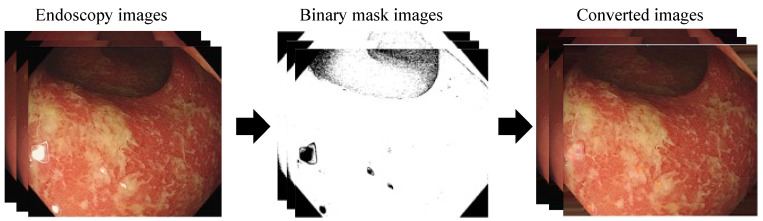
The removal of reflections and dark areas via HSV conversion and inpainting: White: light reflection.

**Figure 6 jpm-13-01584-f006:**
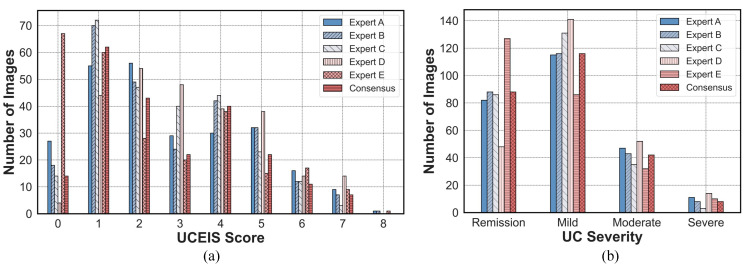
The distribution of (**a**) scores and (**b**) severity from five experts, and consensus data.

**Figure 7 jpm-13-01584-f007:**
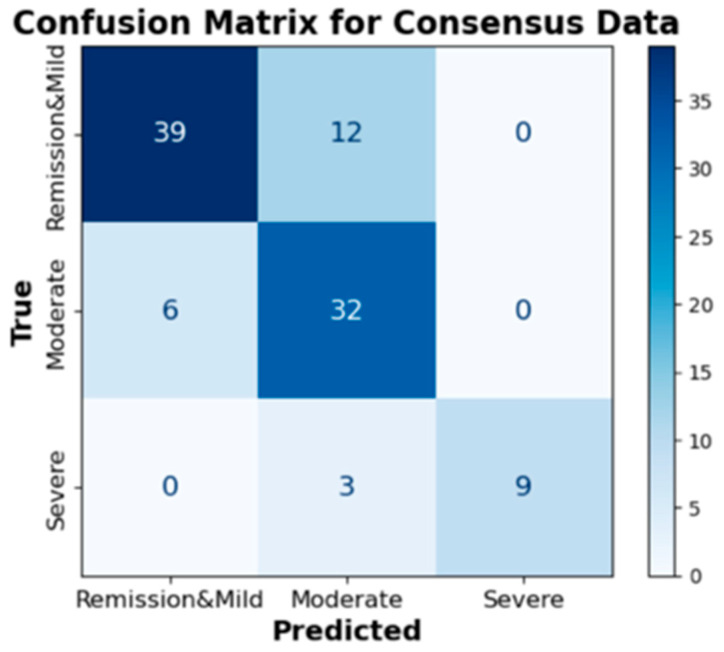
Confusion matrix of the EfficientNetB0 model for consensus data.

**Figure 8 jpm-13-01584-f008:**
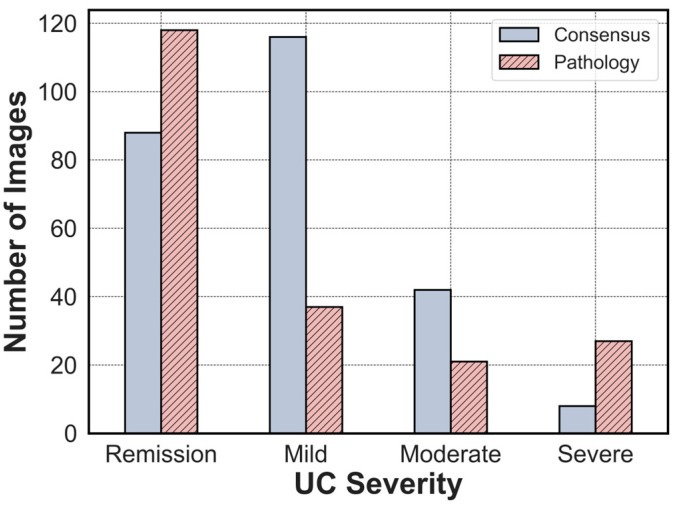
The distribution of severity in consensus and pathology data.

**Table 1 jpm-13-01584-t001:** Demographics and images.

Index		Data
Sex, *n*	Male	57
	Female	58
Age, years	Mean (range)	46 (19–78)
	Median	44
Images	Sampling date	06/2019–02/2021
	Number of images (remission/mild, moderate, severe)	254 (204, 42, 8)

**Table 2 jpm-13-01584-t002:** Distribution of Images for Deep Learning: Training vs. Test Set.

	Severity		
	Remission/Mild	Moderate	Severe
Training set	218	154	48
Testing set	51	38	12
Total	269	192	60

**Table 3 jpm-13-01584-t003:** Interpretation of intraclass correlation coefficients (ICC).

ICC	Level of Agreement
0.9–1.0	Excellent
0.75–0.9	Good
0.5–0.75	Moderate
<0.5	Poor

**Table 4 jpm-13-01584-t004:** Interpretation of kappa index.

Kappa	Level of Agreement
1.00	Perfect
0.81–0.99	Near perfect
0.61–0.80	Substantial
0.41–0.60	Moderate
0.21–0.40	Fair
0.10–0.20	Slight
0	Equivalent to chance

**Table 5 jpm-13-01584-t005:** Performance of deep learning networks on consensus data.

Model	Accuracy	F1 Score	Recall	Precision
EfficientNetB0	0.7920	0.8125	0.7647	0.8666
MobileNetV3Large	0.7473	0.7415	0.7473	0.7686
ResNet50	0.7473	0.7302	0.7473	0.7704
VGG16	0.7363	0.7328	0.7363	0.7422
EfficientNetB7	0.7033	0.6981	0.7033	0.7308
DenseNet121	0.6923	0.6671	0.6923	0.7621
InceptionV3	0.6813	0.6788	0.6813	0.6846
VGG19	0.6813	0.6517	0.6813	0.6502
DenseNet201	0.6374	0.5898	0.6374	0.5736
Xception	0.6044	0.5765	0.6044	0.7029
MobileNetV2	0.5934	0.5864	0.5934	0.5808
ResNet152V2	0 5495	0.5041	0.5495	0.5335
ResNet50V2	0.5385	0.5415	0.5385	0.6988

**Table 6 jpm-13-01584-t006:** The number of pass/fail images among 50 test images with six network models.

	Consensus Model	Expert Models				
		A	B	C	D	E
Pass	46	40	39	44	28	32
Fail	4	10	11	16	22	18

## Data Availability

The data presented in this study are available on request from the corresponding author.

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
