# Peer review of "Using a Deep Learning Model to Address Interobserver Variability in the Evaluation of Ulcerative Colitis (UC) Severity"

_jpm, 2023, doi:10.3390/jpm13111584_

Round 1
Reviewer 1 Report
Comments and Suggestions for Authors
My Comments are as follows:
Dataset:
dataset in my opinion is very limited and it contains only 50 images, it should be justified properly or dataset should be increased.
Comparison:
a comparison table should be added with previously published articles that how this article is even required. Table5 is only limited to your own tests.
Author Response
- Dataset:
dataset in my opinion is very limited and it contains only 50 images, it should be justified properly or dataset should be increased.
(Answer) Thank you for your valuable feedback regarding the size of our dataset. We would like to clarify our dataset composition and provide context.
While it might seem that the dataset consisting of 52 images is limited, these images were specifically chosen as they align with the primary objectives of our study. Moreover, we incorporated images from the HyperKvasir database to augment our dataset. Out of a total of 851 images from the HyperKvasir database, only 262 images were deemed appropriate for our study based on endoscopic findings. Furthermore, these images were assessed by five experts, and only those that met the consensus criteria were included in the training dataset.
By using this carefully curated dataset, we aimed to ensure a comprehensive and accurate training environment for our deep learning model, emphasizing the deliberate and purposeful selection of data for our research.
L 145 – 154: Our initial dataset consisted of endoscopic images split into training and test sets at an approximate ratio of 8:2 (Table 2). To further enhance and generalize our deep learning model, we supplemented our dataset with images from the HyperKvasir open dataset, obtained from Bærum Hospital between 2008 and 2016, which boasts a collection of 110,079 images. From this, 10,662 images were labeled across 23 classes of findings [17], with a significant portion related to pathological findings like Barrett’s esophagus, esophagitis, polyps, ulcerative colitis, and hemorrhoids. From the 644 UC images, we carefully selected those with quality endoscopic features. Five experts then assessed these images for severity, resulting in a consensus on 262 images. These were combined with our initial 254 images, culminating in a total dataset of 516 images for deep learning training.
L 225 – 233: We evaluated the performance of 13 deep learning network models of consensus data, which is crucial for ensuring reliability in endoscopic image classification. Table 5 presents the results, including accuracy, F1 score, recall, and precision. The models are listed in descending order based on accuracy. the EfficientNetB0 exhibited the highest overall performance. With an impressive accuracy of 79.20%, coupled with a balanced F1 score of 81.25%, this model demonstrated robust capabilities in the classification tasks. The con-fusion matrix for the EfficientNetB0 model shows that it correctly identified remission and mild 39 times out of 51, and moderate 39 times out of 41 (Figure 7).
- Comparison:
a comparison table should be added with previously published articles that how this article is even required. Table5 is only limited to your own tests.
(Answer) Thank you for your feedback regarding the comparison with previously published articles.
We have indeed prepared a table that collates our research results in relation to prior studies. This comparison not only helps to justify the need for our study but also emphasizes the unique attributes of our research. One of the distinguishing features of our study, when compared to previous works, is the advanced handling of artifacts like light reflections. We have confirmed that controlling such artifacts significantly contributed to the performance enhancement observed in our results.
Furthermore, our research stands out in the quality of data used for the deep learning model. By constructing consensus data based on labels given by five experts, we ensured that only high-quality images were used for training. This meticulous approach allowed us to achieve commendable accuracy even with a relatively limited dataset.
Additionally, we are in the process of expanding our study to a multi-institutional level. This expansion is anticipated to further improve the performance of our system. Therefore, we are confident that subsequent endeavors will yield deep learning models with even greater accuracy and utility.
We have made revisions to the manuscript to ensure that these aspects are prominently highlighted and easily discernible.
We appreciate your insights and are grateful for the guidance you have provided.
L 259 - 274: The comparison with previous studies on the evaluation of UC severity using deep learning is presented in Table 6. One of the notable features of our study is the advanced technology for handling artifacts like light reflections. Thanks to this approach, the performance of our deep learning model has improved. Moreover, the number of images used in our study might be relatively small when compared with other deep learning studies involving endoscopic images. While the volume of data is important in deep learning, we recognized that the use of high-quality data is even more critical. Therefore, we selectively utilized images of the rectum from patients. This is because UC symptoms first appear in the rectum, manifesting most prominently there before gradually spreading to other areas.
Additionally, in the labeling process, we used only data that met certain criteria to minimize inter-observer variability. Still, in situations where data was scarce, we combined our data with the HyperKvasir dataset for deep learning model training. From the HyperKvasir dataset, we selectively curated images that were suitable for endoscopic judgment and of good quality to form a consensus data set. Owing to our meticulous data management, we achieved excellent accuracy even with a relatively small dataset [22,23].

Reviewer 2 Report
Comments and Suggestions for Authors
The current study used an AI model to judge the severity of UC, which is significant and reliable.
1.The training sample size was much smaller with moderate and severe compared with mild/remission.
2.The mild and remission were not seperated in Table 2 but was seperated in Fig 6b.
3.Has it been compared between the model and experts in further images?
Author Response
The current study used an AI model to judge the severity of UC, which is significant and reliable.
1. The training sample size was much smaller with moderate and severe compared with mild/remission.
(Answer) Thank you for your valuable feedback regarding the size of our dataset. We would like to clarify our dataset composition and provide context.
While it might seem that the dataset consisting of 52 images is limited, these images were specifically chosen as they align with the primary objectives of our study. Moreover, we incorporated images from the HyperKvasir database to augment our dataset. Out of a total of 851 images from the HyperKvasir database, only 262 images were deemed appropriate for our study based on endoscopic findings. Furthermore, these images were assessed by five experts, and only those that met the consensus criteria were included in the training dataset.
By using this carefully curated dataset, we aimed to ensure a comprehensive and accurate training environment for our deep learning model, emphasizing the deliberate and purposeful selection of data for our research.
L 145 – 154: Our initial dataset consisted of endoscopic images split into training and test sets at an approximate ratio of 8:2 (Table 2). To further enhance and generalize our deep learning model, we supplemented our dataset with images from the HyperKvasir open dataset, obtained from Bærum Hospital between 2008 and 2016, which boasts a collection of 110,079 images. From this, 10,662 images were labeled across 23 classes of findings [17], with a significant portion related to pathological findings like Barrett’s esophagus, esophagitis, polyps, ulcerative colitis, and hemorrhoids. From the 644 UC images, we carefully selected those with quality endoscopic features. Five experts then assessed these images for severity, resulting in a consensus on 262 images. These were combined with our initial 254 images, culminating in a total dataset of 516 images for deep learning training.
L 225 – 233: We evaluated the performance of 13 deep learning network models of consensus data, which is crucial for ensuring reliability in endoscopic image classification. Table 5 presents the results, including accuracy, F1 score, recall, and precision. The models are listed in descending order based on accuracy. the EfficientNetB0 exhibited the highest overall performance. With an impressive accuracy of 79.20%, coupled with a balanced F1 score of 81.25%, this model demonstrated robust capabilities in the classification tasks. The con-fusion matrix for the EfficientNetB0 model shows that it correctly identified remission and mild 39 times out of 51, and moderate 39 times out of 41 (Figure 7).
2. The mild and remission were not seperated in Table 2 but was seperated in Fig 6b.
(Answer) Thank you for your valuable feedback.
In Table 2, we combined the categories of remission and mild severity to demonstrate the composition of the dataset fed into the deep learning model. The decision to merge these categories arises from a clinical rationale and is guided by treatment objectives. Specifically, from a therapeutic standpoint, patients either in remission or exhibiting mild symptoms are typically not prioritized for aggressive intervention. By grouping these categories, we aim to establish a clinically meaningful distinction in the management of the condition. This strategy ensures that the model's predictions resonate more accurately with the clinical considerations that drive treatment decisions, potentially amplifying the model’s relevance in real-world clinical scenarios.
On the other hand, in Fig 6b, we segregated remission and mild categories to analyze the extent of interobserver variation when assessing UC severity. After statistically measuring the interobserver variation, we constructed the dataset for the deep learning model based on data where a majority agreed upon its interpretation. The decision to present the combined remission and mild categories in Table 2 was driven by this clinical rationale.
We recognize that the contrasting representations might be a source of confusion. To clarify, we will elaborate on the reasons and context behind these differences more comprehensively in the main text of the paper.
3. Has it been compared between the model and experts in further images?
(Answer) Thank you for your inquiry.
To enrich our study, we integrated an additional dataset, namely the "Hyperkvasir" dataset, into our existing data collection. Subsequently, five experts were engaged to provide labels on this augmented dataset. Using our deep learning model, a comparative analysis was conducted to evaluate its performance on this enhanced data compilation. The results stemming from the incorporation of the Hyperkvasir dataset can be found in Table 5, as referenced in comment #1. We have also made necessary amendments in the manuscript to reflect the outcomes and implications of adding this new dataset.
We trust that these modifications offer greater clarity and depth to our research.
